# Classification of Snow Cover Persistence across China

**Hongxing Li [1,2], Xinyue Zhong [1,*], Lei Zheng [3], Xiaohua Hao [1] , Jian Wang [1] and Juan Zhang [4]**

1   Northwest Institute of Eco-Environment and Resources, Chinese Academy of Sciences,
    Lanzhou 730000, China; lihongxing@lzb.ac.cn (H.L.); haoxh@lzb.ac.cn (X.H.); wjian@lzb.ac.cn (J.W.)
2   National Cryosphere Desert Data Center, Lanzhou 730000, China
3   School of Geospatial Engineering and Science, Sun Yat-sen University, Guangzhou 510275, China;
    zhenglei6@mail.sysu.edu.cn
4   Meteorological Institute of Qinghai Province, Xining 810001, China; babyzj@lzb.ac.cn
*   Correspondence: xyzhong@lzb.ac.cn

**Abstract:** In this study, we classified the variability in snow cover persistence across China by using a novel method; continuous snow cover days and variability of snow cover were used as the evaluation indicators based on a long-term Advanced Very High Resolution Radiometer (AVHRR) snow cover extent (SCE) product. The product has been generated by the snow research team in the Northwest Institute of Eco-Environment and Resources (NIEER), Chinese Academy of Sciences. There were obvious differences in snow cover classification in three snow cover areas (northern Xinjiang, northeast China, and the Tibetan Plateau): northern Xinjiang was dominated by persistent snow cover, most regions of northeast China were covered by persistent and periodic variable snow cover. There was the most abundant snow cover classification in the Tibetan Plateau. The extents of persistent and periodic variable snow cover were gradually shrinking due to rising temperatures and decreasing snowfall during 1981–2019. In contrast, non-periodic variable snow cover areas increased significantly. This method takes into account the stability, continuity, and variability of snow cover, and better captures the characteristics and changes of snow cover across China. Based on our research, we found that snow disasters in ephemeral-type (belong to non-periodic variable snow cover) regions cannot be well prevented because of the unfixed snow cover timing. Therefore, we recommend that monitoring and forecasting of snow cover in these snow cover regions should be strengthened.

**Keywords:** snow cover classification; interdecadal variation; snow disaster; China

## 1. Introduction

Snow cover is an indicator of climate change. It plays an important role in the surface energy budget [1], the surface-atmosphere interaction [2], and the hydrological processes [3]. The distribution and change of snow cover are inhomogeneous across China [4–6]. Regional water supply, agriculture, and oasis irrigation are highly dependent on local snow conditions. In addition, snow disasters such as avalanches, blizzards, and blowing snow have severely affected human activities. Thus, it is necessary to classify snow cover based on its stability and persistence across China and identify the snow cover characteristics and its effects in different regions.

As early as 1946, Formozov [7] first proposed to use vegetation and ecological regions as criteria to divide seasonal snow cover of the former Soviet Union. He divided snow cover into four types: tundra, forest, steppes and desert, and mountain snow. Since then, more and more research focused on snow cover classification, but there was no standard formed. The indicators of snow cover classification were mainly divided into physical parameters and processes of snow cover (snow grain size, snow density, snow hardness, snow depth, snow water content, snow cover duration, snow temperature, etc.) [8–11], and the external environment (vegetation, ecological environment, climate, etc.) [12–16]. On this basis,

Sturm et al. proposed a new snow cover classification according to snow properties (snow depth, snow density, snow layer, etc.) and climatic conditions (temperature, precipitation, wind speed, etc.) [17]. Snow cover is divided into the tundra, taiga, alpine, maritime, prairie, and ephemeral snow. This method is widely recognized and applied. They revisited the snow classification system and presented a revised classification using new datasets and methods in 2021 [18]. Royer et al. proposed an updated snow cover classification in three classes, which was more appropriate to northeastern Canada than the general classification commonly used [19]. In addition, based on persistence, snow cover is divided into persistent and variable snow cover. The number of continuous snow cover days equals to or greater than 30 days as a criterion [20].

The spatial and temporal distributions of snow depth and snow water equivalent (SWE) across China are greatly affected by the local climate, topography, and vegetation [6,21,22]. Thus, compared with snow depth, snow cover timing is more representative of reflecting snow cover distribution. On the other hand, compared to other snow cover parameters (such as snow depth, snow density, and SWE), surface energy budget, soil thermal conditions, river runoff, ecosystem, and the occurrence of cryospheric disasters are more significantly affected by the length and changes of snow cover timing [23–26]. The spatiotemporal resolution of snow cover days (SCD) from remote sensing is also higher than others. Therefore, SCD is more suitable to be used as an index for snow cover classification based on stability and persistence in China.

Snow cover classifying by SCD based on persistence was first proposed by Li and Mi [27] in China. The interannual mean accumulated snow cover days (ASCD) were used as a criterion for snow cover classification. Snow cover was divided into persistent snow cover (ASCD $\geq$ 60), periodic variable snow cover (10 $\leq$ ASCD < 60), and non-periodic variable snow cover (0 < ASCD < 10) according to the climatology of ASCD across China. Due to the limitation of observed data, the continuous characteristics of snow cover and the variations in snow cover timing were not fully considered in this method. Then, He and Li [28] established a new indicator combining the climatology and the interannual variability of ASCD to classify snow cover with observations (1951–2004) and remote sensing data (1980–2004) in western China. However, this method also ignored the continuity of snow cover. In addition, snow depths derived from the scanning multichannel microwave radiometer (SMMR) and the special sensor microwave imager (SSM/I) were used to identify SCD in their research, while the accuracy of the result is always affected by low spatial resolution (25 $\times$ 25 km), underlying surface conditions, and algorithms. Although Zhang and Zhong's method used the maximum continuous snow cover days (CSCD) obtained from ground-based data to classify snow cover and fill the gap of snow cover persistence in previous studies, the variations in snow cover timing were not mentioned [29]. Furthermore, regional deficiency of in situ measurement is also an important issue that cannot be solved. Thus, it is necessary to establish a novel classification index with a high resolution of remote sensing data, considering the snow cover continuity and variations in snow cover timing, and classify snow cover across China.

In this study, we classified the variability in snow cover persistence with a recently developed method of CSCD across China, which we used to clarify snow cover characteristics in different snow cover areas. Section 2 describes data sources, definitions of snow cover timing, classification criteria, and the anomaly calculation method. Section 3 details the results of snow cover classification and its interdecadal variations. We compared the differences between the recent and old methods, as well as the disaster-causing and beneficial characteristics of snow cover areas across China. Finally, key findings are summarized in Section 4. We hope that the recent snow cover classification will provide basic data for climate change, hydrology, the ecosystem, and disaster prevention and mitigation.

## 2. Data and Methodology

### 2.1. Data Sources

Snow cover extent (SCE) data were used to identify SCD. A new SCE product, used in this study, was generated by the snow research team in the Northwest Institute of Eco-Environment and Resources (NIEER), Chinese Academy of Sciences. The product dataset of snow phenology in China is based on the Advanced Very High Resolution Radiometer (AVHRR) from 1981 to 2020 [30]. This is a daily SCE product with a spatial resolution of 5 km. It is a completely gap-free SCE product, produced through a series of processes, such as quality control, cloud detection, snow discrimination, and gap-filling. This product used a multi-level decision tree algorithm to discriminate the cloud and snow cover and improved the gap-filling technique. The NOAA CDR of AVHRR Surface Reflectance, version 4 (AVHRR SR V4) was used for the basic input data. Ground-based measurements of snow depth from meteorological stations and higher spatial resolution SCE maps derived from Landsat-5 Thematic Mapper (TM) were the validated data of the new SCE product. Two groups of Landsat-5 TM maps were used across China from 1985 to 2013. The first group was used as the "true" values to acquire the training data of AVHRR surface reflectance. TM snow maps were produced by the improved "SNOMAP" algorithm developed by Chen et al. [31] for the snow season (beginning on 1 November to 31 March of the following year). Each map contained three classes (snow, non-snow, and cloud). The second group of maps was used as ground truth values to evaluate the AVHRR SCE product. To ensure these products' accuracy, we also checked them again by our visual interpretation. On average, their accuracies were larger than 99% compared to the results of our visual interpretation. The overall accuracy reference to ground snow depth measurements and Landsat-5 was 87.4% and 87.3%, respectively, which were significantly higher than other current AVHRR products. A more detailed introduction to the product is provided by Hao et al. [30].

To analyze the influencing factors of the variations in snow cover classification, daily air temperature, precipitation, and snow depth observations at 452 meteorological stations across China during 1981–2020 were collected from the National Meteorological Information Center (NMIC, China Meteorological Administration) (Figure 1). Daily precipitation data were divided into solid and liquid fractions (represented by the solid fraction $S_{rat}$) according to the daily mean air temperature ($\overline{T}$, °C) as [32]:

$$S_{rat} = \begin{cases} 1.0 & for\ T_{mean} \leq -2.0\,°C, \\ 0.0 & for\ T_{mean} \geq +2.0\,°C, \\ 1.0 - 0.25(T_{mean} + 2.0) & for\ -2.0\,°C < T_{mean} < +2.0\,°C. \end{cases} \tag{1}$$

Daily snowfall was calculated as the product of daily precipitation and the daily $S_{rat}$ value.

### 2.2. Definitions and Methods

We defined a snow year as the period from 1 September to 31 August of the following year. SCD was used to estimate the snow season length (the accumulative number of days between the onset and termination of snow cover), ASCD, CSCD, and the variations in snow cover. We defined the variables for each pixel with snow cover:

(1)    SCD: snow cover day, any day with snow cover in each pixel for each snow year;
(2)    SSL: snow season length, the number of days from the first to the last snow cover day in each snow year;
(3)    ASCD: the accumulated snow cover days in each snow year;
(4)    CSCD: the continuous snow cover days in each snow year;
(5)    RAL: the ratio of ASCD to SSL, reflecting the variability of snow cover; and
(6)    Long-term mean annual ASCD, CSCD, and RAL: averaged annual values in the snow years for 1981–2019.

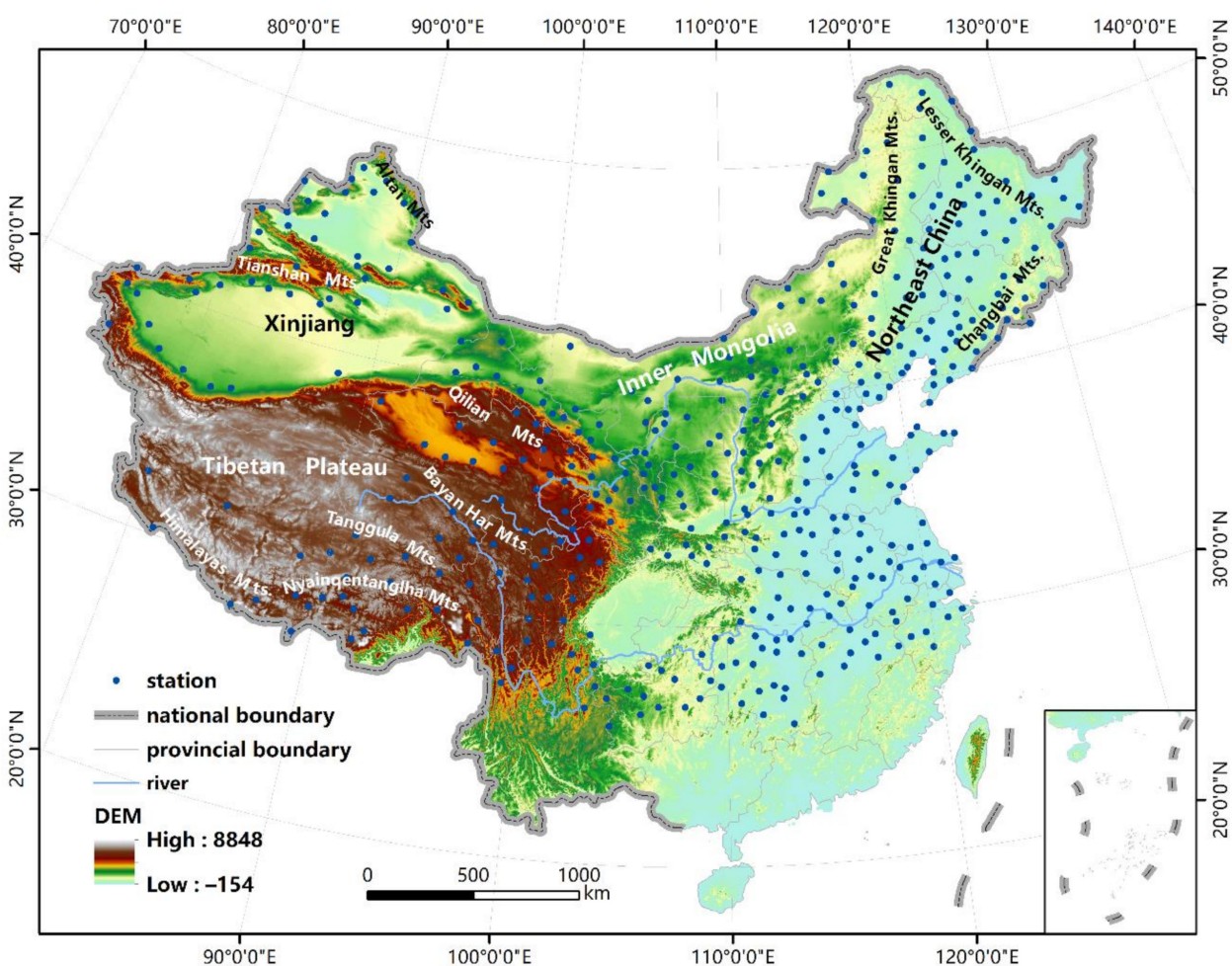

**Figure 1.** Geographical locations of 452 meteorological stations across China.

To minimize random and systematic errors in the dataset, we implemented a consistent quality control according to the assessment criteria. As the World Meteorological Organization's approach to calculating anomalies is based on 30-year climate normal periods [33], we used the period 1981–2010 as our normal period. To ensure data continuity, each pixel with less than 10 years of snow cover from 1981 to 2010 was excluded.

Snow cover was divided into three types: persistent snow, periodic variable snow, and non-periodic variable snow. The long-term mean annual maximum CSCD (MCSCD) and long-term mean annual RAL were the classification index. Regions, where MCSCD was equal to or greater than 30 days, were classified as the persistent snow cover areas [20]. Periodic variable snow cover areas were the regions with $10 \leq$ MCSCD $< 30$ d. The non-periodic variable snow cover (0 < MCSCD < 10 d) was divided into two classifications: fixed-type (long-term mean annual RAL $\geq 0.14$) and ephemeral-type (long-term mean annual RAL < 0.14). MCSCD = 10 was used as a criterion for dividing periodic and non-periodic variable snow cover, and was based on the long-term distribution characteristics of SCD across China [27,29]. Furthermore, 0.14 was the median of the long-term mean annual RAL for all pixels belonging to the non-periodic variable snow cover across China. Furthermore, fixed snow cover indicates that there is snow cover in these areas during a fixed period of each snow cover year, while the ephemeral snow cover means that snow cover timing is fluctuating. It is difficult to prevent once extreme snowfall or a snow disaster occurs in these areas. All cartography of snow classifications and variations in snow cover were calculated and plotted in ArcGIS 10.7.

We used the anomaly method to partly overcome problems arising from different station data availability periods and absolute background conditions (e.g., elevation). We calculated anomalies of the annual air temperature, snowfall, and SCD values relative to the climate normal period (1981–2010) for each station, then averaged the station-level anomalies into anomalies for all in China. The Student's *t*-test was used to evaluate the statistical significance of the changes in air temperature, snowfall, and SCD. We set the significance level of this study at $\alpha = 0.05$.

## 3. Results and Discussion

### 3.1. Climatology of Snow Cover Classification

The recently developed snow cover classification across China from 1981 to 2019 indicated that persistent snow cover extent was about $190.12 \times 10^4$ km$^2$ (19.8% of China) (Table 1) and mainly located in most areas of the northern Xinjiang Uygur Autonomous Region, the Great Khingan Mountains, the Lesser Khingan Mountains, Sanjiang Plain, Songnen Plain, some areas of the Changbai Mountains, and the high-mountain Tibetan Plateau (including most of the Himalayas and Nyainqentanglha Mountains, some parts of Tanggula, Bayan Har, and Qilian Mountains) (Figure 2, blue color). The long-term mean annual MCSCD was more than 30 days in these regions.

**Table 1.** Extents of snow cover classification and their interdecadal relative change rates.

| Snow Classification | Snow Cover Extent ($\times 10^4$ km$^2$) | | | | | Rate of Change in Extent (%) | | |
|---|---|---|---|---|---|---|---|---|
| | 1981–2019 | 1980s | 1990s | 2000s | 2010s | 1990s | 2000s | 2010s |
| Persistent snow cover | 190.12 | 190.32 | 187.93 | 197.64 | 190.91 | −1.26 | 5.17 | −3.41 |
| Periodic variable snow cover | 206.48 | 205.19 | 209.97 | 210.74 | 195.97 | 2.33 | 0.37 | −7.01 |
| Fixed-type | 419.71 | 420.34 | 414.5 | 397.35 | 413.67 | −1.39 | −4.14 | 4.11 |
| Ephemeral-type | 69.92 | 70.38 | 73.82 | 80.5 | 85.67 | 4.89 | 9.05 | 6.42 |

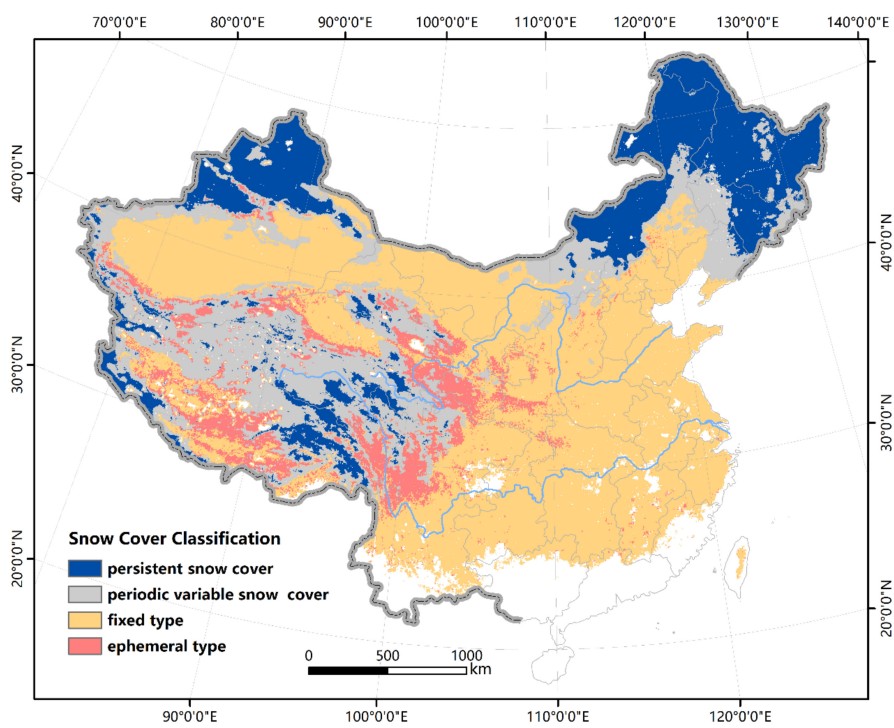

**Figure 2.** Distributions of the recently developed snow cover classification based on the persistence across China during 1981–2019.

Periodic variable snow cover areas (~206.48 × 10$^4$ km$^2$, 21.51% of China) were found in most of the Tibetan Plateau, the southern Tianshan Mountains, central Inner Mongolia, the eastern Changbai Mountains, western Jilin Province, and eastern Liaoning Province (Figure 2, gray color), where the long-term mean annual MCSCD was more than 10 days but less than 30 days. Different from the other two major snow cover areas across China, there is a special non-periodic variable snow cover in the Tibetan Plateau, namely the ephemeral-type (approximately 69.92 × 10$^4$ km$^2$, red color in Figure 2). It accounts for 7.28% of China. This snow cover classification showed that the long-term mean annual MCSCD was no more than 10 days and RAL was less than 0.14. The low RAL indicated that, although there was the early onset and late end of snow cover, ASCD was very short in the eastern and southern hinterland of the Tibetan Plateau during the whole SSL. In these regions, snow cover timing is not fixed, and it is difficult to prevent snow disasters once extreme snowfall occurs or a large amount of snow cover accumulates. Therefore, we need to focus on these areas. Most of the other snow cover areas across China were fixed-type (Figure 2, yellow color), which belonged to non-periodic variable snow cover (0 < MCSCD < 10 d, RAL ≥ 0.14) and had the largest extent of all snow cover classification, about 419.71 × 10$^4$ km$^2$ (43.72% of China). Snow cover accumulated within a fixed and short period in these regions. The snow-free areas were mainly located in the south of 28° N between 118–121° E, the south of 26° N between 106–117° E, and the south of 25° N between 99–104° E, about 7.69% of China.

The biggest differences between the recently developed and traditional (long-term mean annual ASCD as a criterion, Figure 3) classifications were snow cover types in the Tibetan Plateau and the extent of periodic variable snow cover across China. The results showed that compared to the traditional method, the extents of both persistent and periodic variable snow cover classified by the recent method were significantly decreased, in contrast, the non-periodic variable snow cover extent increased across China, especially in the Tibetan Plateau. Snow depth observations from meteorological stations showed that snow cover disappeared completely within a few days or even a few hours for one snowfall event in some regions of the Tibetan Plateau and northern China. However, these areas were classified as persistent or periodic variable snow cover by the traditional method. For snow cover classification based on persistence, it is necessary to consider the continuity and stable characteristics of snow cover. Therefore, the recent classification is more in line with the snow cover conditions and captures the changes of snow characteristics across China. Furthermore, the recent method identified potential snow risk areas (ephemeral-type) in the Tibetan Plateau, which can provide important information and a basis to focus on snow cover changes in these areas.

Using the same criterion (long-term mean annual ASCD), to compare snow cover classifications before and after the 1980s [27], indicated that the spatial pattern of snow cover classification across China has changed dramatically. The extent of persistent and non-periodic variable snow cover increased in northern Xinjiang. On the contrary, periodic variable snow cover in this region decreased significantly. The extent of inverted U-shaped distributions of persistent snow cover in northeast China and central and eastern Inner Mongolia gradually moved northward. In the Tibetan Plateau, the snow cover distribution was dominated by persistent snow cover before 1980, which had changed to the periodic variable snow cover classification in the past 40 years. The persistent snow cover was mainly distributed in the high mountains. The rest of China showed a significant reduction in the extent of periodic variable snow cover and an obvious expansion in non-periodic variable snow cover. In addition, the snow-free areas had also increased. There was a long-term significant increasing trend in air temperature with the rate of 0.3 °C/decade (Figure 4). Although the changes were not statistically significant, both snowfall and SCD were decreasing. The interannual variation of snowfall was relatively smooth, which was not synchronized with the variation trend of SCD. However, the changes in SCD and air temperature presented a corresponding opposite trend. It indicated that the variations in snow cover classification areas may be more closely related to a higher temperature than

decreased snowfall. Light-absorbing impurity deposited on snow is also one of the factors that accelerate snowmelt [34,35].

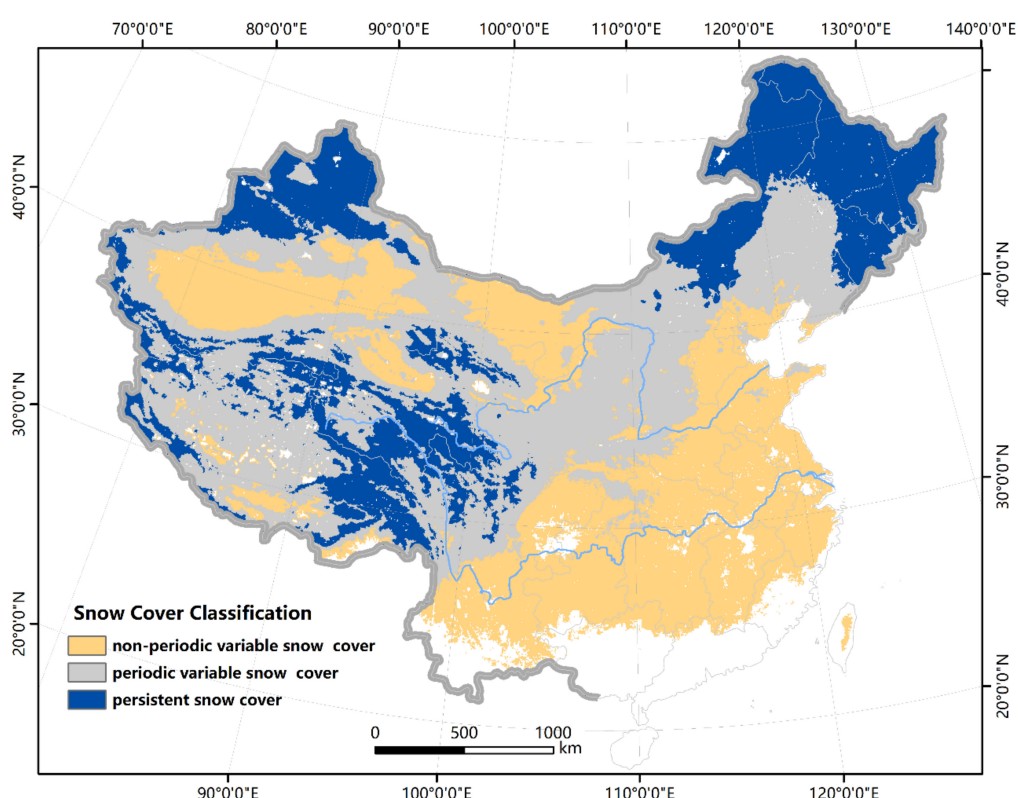

**Figure 3.** Distributions of the traditional snow cover classification across China during 1981–2019.

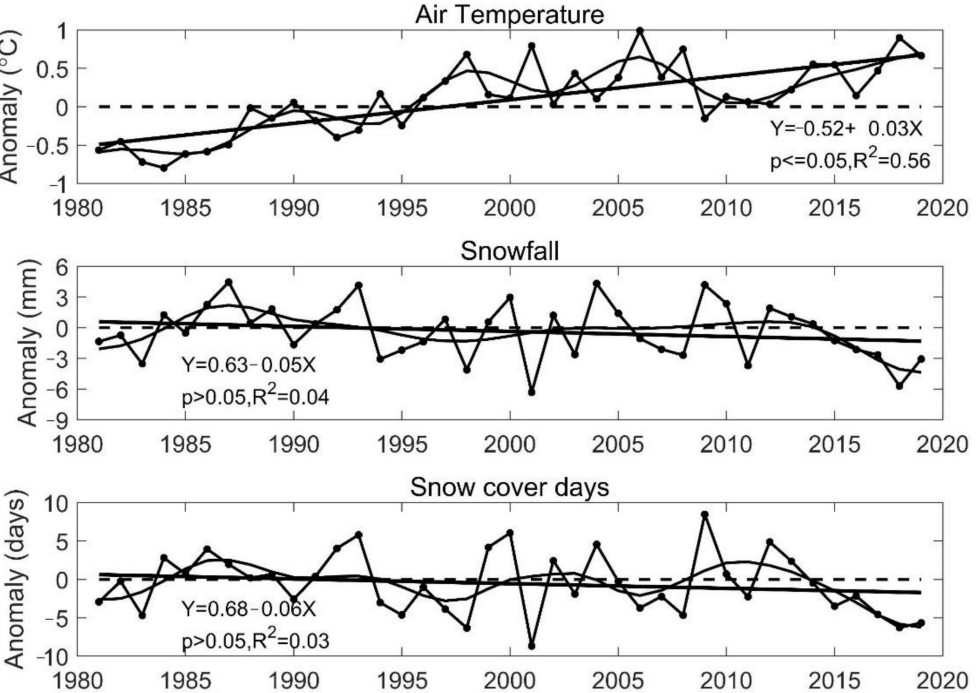

**Figure 4.** Interannual variations of annual mean air temperature, snowfall, and snow cover days from 1981 to 2019 with respect to the 1981–2010 mean across China.

### 3.2. Interdecadal Variations in Snow Cover Classification

The interdecadal variations in snow cover classification showed that the largest extent of persistent snow cover appeared in the 2000s (~197.64 × 10$^4$ km$^2$), and the smallest extent was in the 1990s (about 187.93 × 10$^4$ km$^2$) (Table 1, Figure 5). Periodic variable snow cover extent increased from the 1980s to the 2000s and then decreased significantly with the relative change rate of −7.01% during the 2010s compared with the 2000s. Fixed-type in the non-periodic variable snow cover accounts for the maximum extent of snow cover classification across China. Furthermore, contrary to the interdecadal variation in periodic variable snow cover, the snow cover extent of the fixed-type represented a gradually decreasing trend from the 1980s to the 2000s. After 2010, the extent of the periodic variable snow cover decreased significantly, most of which changed to non-periodic variable snow cover, and the fixed-type extent increased obviously by 4.11%. The ephemeral-type of the extent of snow cover classification gradually increased, ranging from 70.38 × 10$^4$ km$^2$ in the 1980s to 85.67 × 10$^4$ km$^2$ in the 2010s (~3.82 × 10$^4$ km$^2$/decade). In addition to the ephemeral-type in the Tibetan Plateau, the extent of this snow cover classification in the southern margin of snow cover areas across China was also increasing significantly during the recent 40 years.

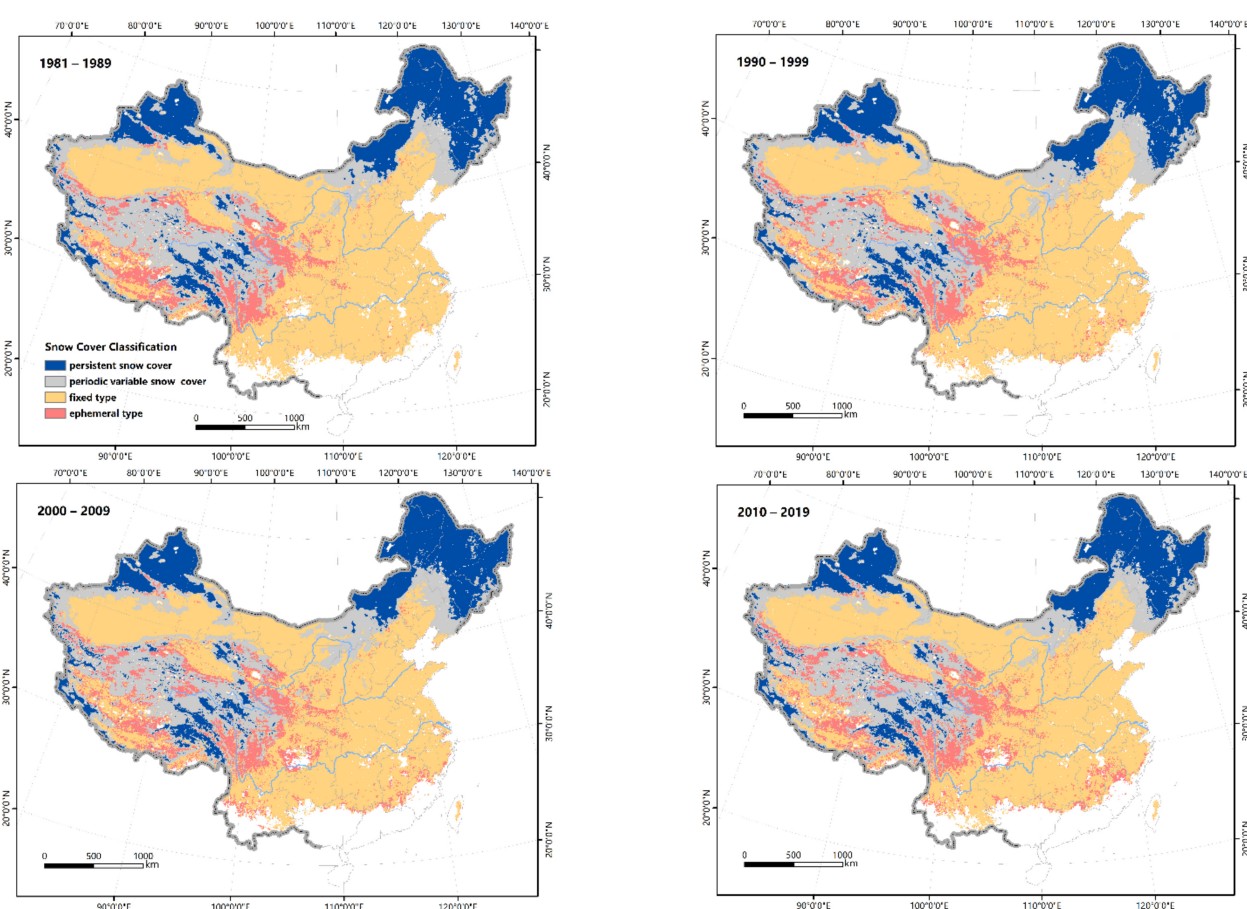

**Figure 5.** Interdecadal distributions of the recently developed snow cover classification across China from the 1980s to the 2010s.

The distribution changes of snow cover classification in the three main snow cover areas (northern Xinjiang, northeast China, and the Tibetan Plateau) showed that northern Xinjiang was dominated by persistent snow cover and the changes of snow cover extent were not obvious from the 1980s to the 2010s (Figure 6). However, at the junction of periodic variable snow cover and fixed-type snow cover, the extent of periodic variable snow cover

increased from the 1980s to the 2000s, while gradually decreasing in the 2010s. Compared with the 1980s, some regions belonging to persistent snow cover changed to periodic variable snow cover, and periodic variable snow cover transformed into ephemeral-type snow cover during the 1990s in northeast China. Then, the extents of both persistent and periodic variable snow cover in northeast China reached their maximum during the 2000s. After that, more persistent snow cover decreased to periodic variable snow cover, and increasing fixed-type snow cover appeared in these regions. Since the 1980s, persistent snow cover extent had gradually declined, which changed to periodic variable snow cover in the Tibetan Plateau. Furthermore, ephemeral snow cover had expanded significantly from the 1990s to the 2000s. In the northern edge of the Tibetan Plateau, periodic variable snow cover and fixed-type snow cover changed alternately during the 1980s–2010s.

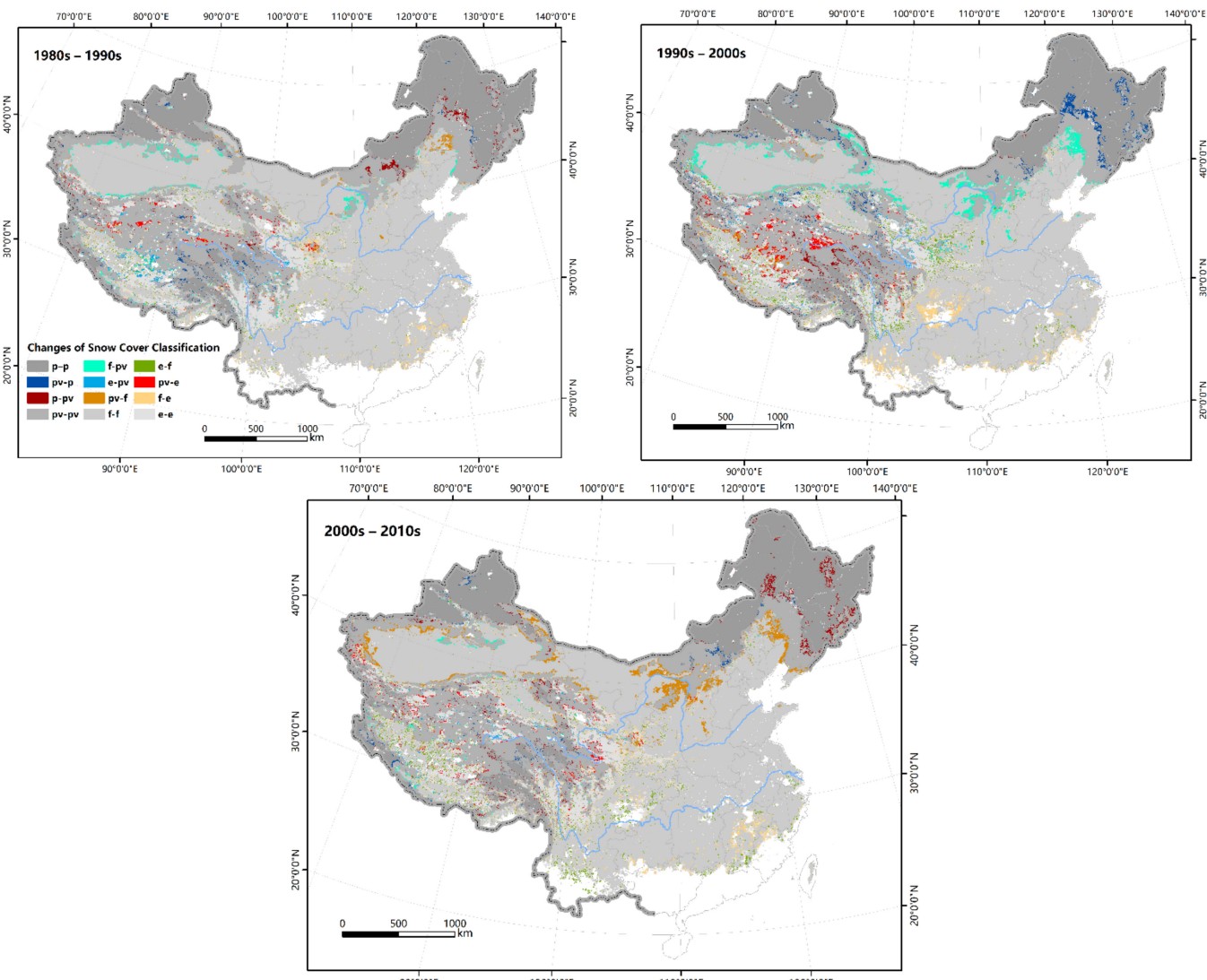

**Figure 6.** Interdecadal variations in the recently developed snow cover classification across China from the 1980s to the 1990s (**left**), from the 1990s to the 2000s (**right**), and from the 2000s to the 2010s (**bottom**). The colors represent the changes in different types. The alphabet is the first letter of the snow cover classification.

Interdecadal variations in snow cover classification revealed the spatial heterogeneity in the changes of snow cover timing. In our previous studies, we observed statistically significant long-term trends of delayed onset and advanced disappearance of snow cover in most areas of China [5]. Snow depth represented an increase in northern Xinjiang and

northeast China while decreasing in other regions [4]. The relationships between SCD, air temperature, and snowfall were analyzed. Significant negative correlations between SCD and air temperature were observed in the northern Tianshan Mountains, northeast China, the TP, and south of the Yangtze River. There were significant positive correlations between SCD and snowfall in the northern Tianshan Mountains and northeast China. Although SCD was highly correlated with air temperature in these two regions, mean air temperatures were below 0 °C during the cold season and an increasing temperature could not have directly resulted in SCD change. The strong correlation between SCD and snowfall implies that increased snowfall led to the observed increase in SCD. Therefore, this could explain the extent of persistent snow cover remaining unchanged or increasing in parts of northern Xinjiang and northeast China. Due to the low and discontinuous snowfall in the Tibetan Plateau, snow cover is more sensitive to the changes in air temperature. SCD and snow depth decreased significantly as air temperature increased. As a result, persistent and periodic variable snow cover shifts to non-periodic variable snow cover in the Tibetan Plateau. The distribution and variations of snow cover are complex, and are not controlled by a single climate factor. In addition to air temperature and snowfall, snow cover is also affected by local climatic conditions, monsoons, atmospheric circulation, etc. It is necessary to further analyze the reasons for the interdecadal variations in snow cover classification.

*3.3. Snow Cover Distribution in Sub-Regions for Snow Cover Classification*

In the three main snow cover regions across China, the spatial patterns of snow cover classification were different due to the large differences in climatology and variations in snow properties. Long-term mean annual ASCDs in northern Xinjiang were more than 60 days, especially in the Altai Mountains and the northern Tianshan Mountains, snow cover accumulated for more than half a year (Figure 7). Long-term mean and maximum snow depths in northern Xinjiang were also the largest among the three sub-regions. The maximum snow depths were more than 20 cm in northern Xinjiang [4]. There were significant increasing trends of annual mean and maximum snow depth in northern Xinjiang during 1966–2012 [4]. Therefore, the interdecadal variation in the extent of persistent snow cover was not evident.

The areas with longer ASCD (>60 days) and deeper snow depth (>3 cm) in northeast China and Inner Mongolia showed an inverted U-shaped distribution (Figure 7) [35], which was very similar to the distribution of persistent snow cover in this region (Figures 2 and 5). The maximum ASCD and snow depth were found in the Great Khingan Mountains and the Lesser Khingan Mountains. Studies showed that both snow depth and RAL had increased significantly in northeast China and Inner Mongolia [5,36]. This increasing RAL indicated that the snow accumulation time was becoming increasingly constant.

Similar to the persistent snow cover distribution, the larger values of ASCD and snow depth in the Tibetan Plateau were mainly located in high-mountain regions. The long-term mean annual ASCD was more than 180 days (Figure 7) and snow depth was less than 14 cm [37]. However, both of them tended to decrease throughout the Tibetan Plateau. Based on this, persistent snow cover gradually changed into periodic variable snow cover or even non-periodic variable snow cover. Due to the irregular snow cover timing, ephemeral snow cover increased obviously.

Snow cover is an important freshwater resource and water supply in arid and semi-arid regions across China, while too deep snow depth and long continuous snow accumulation can easily lead to snow disasters in pastoral areas. Meanwhile, snow disasters also include avalanches, blowing snow, a disaster of low temperature, persistent rain, snow, and ice storms, etc., which seriously affect agriculture, animal husbandry, transportation, and power systems. Snow disasters in pastoral areas across China mainly occur in the Altai Mountains, Tianshan Mountains, Ili River Valley, and Xilingol League in Inner Mongolia, which all belong to persistent snow cover areas [38]. There are thick snow depths, long CSCDs, and fixed snow cover timing (large RAL) in these regions. Snow disasters in pastoral areas also occur in ephemeral snow cover regions in the Tibetan Plateau [38]. The

snow cover accumulation timing in these areas is not fixed. Therefore, once a snow disaster occurs, it is difficult to prevent and needs to be paid attention to.

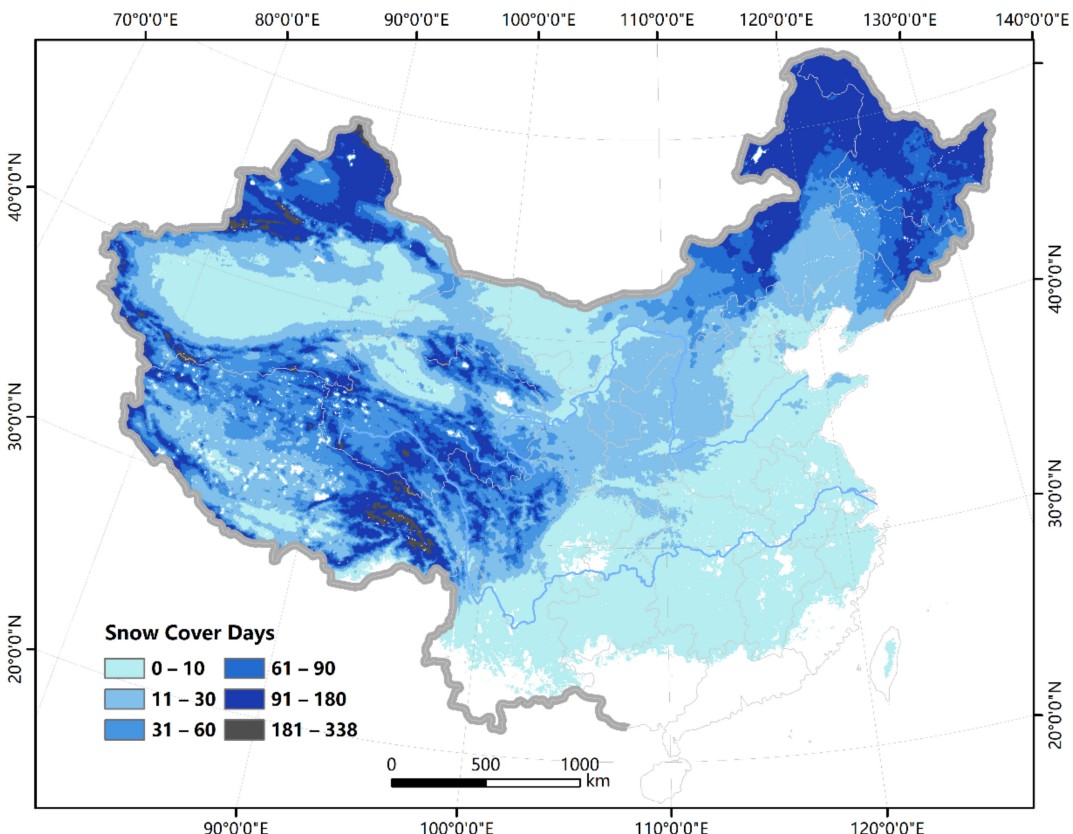

**Figure 7.** Distributions of the long-term mean annual ASCDs across China during 1981–2019.

Avalanches mostly occur in mountainous areas (the Altai Mountains, the Tianshan Mountains, the eastern and southeastern mountainous areas in the Tibetan Plateau) with complex terrain [38]. There is a thicker depth hoar and more water content in the snow profile, especially in spring, leading to the destruction of snow cover stability. Blowing snow is affected by climatic factors, topography, and geographical location. It occurs occasionally in the high-mountain areas of northwest and southwest China, the Inner Mongolia Plateau, and the Songliao Plain of northeast China. Avalanches and blowing snow are usually observed in persistent and periodic variable snow cover areas. Although these two types of snow disasters are more harmful, special monitoring and forecasting have been carried out based on the relatively fixed occurrence time of snowfall and snow cover accumulation.

The disaster of low temperature, persistent rain, snow, and ice storms mainly appear in the southern regions of China, which belong to ephemeral snow cover areas. It is difficult to prevent snow disasters in these areas because of the unfixed snow cover timing. Once the low temperature, persistent rain, snow, and ice storms occur, they will have a huge impact on the local transportation, power system, and residents' lives.

In summary, snow disasters in persistent and periodic variable snow cover areas across China have relatively fixed occurrence timing and locations. The prevention of snow disasters in these areas should focus on monitoring and forecasting snow depth and CSCD. However, it is not easy to prevent snow disasters in non-periodic variable snow cover areas, especially in ephemeral-type regions. Thus, we should pay more attention to local meteorological information and carry out forecast evaluations.

Previous studies revealed that snow cover extent and SCD decreased in 78% of global mountain areas, while few areas showed positive changes [39]. Similar results were also

presented over northern Eurasia from 1966 to 2007 [40]. The forecast future trend in snow cover indicated there would be a 40% decrease in SCD [41]. The SCD variations have implications for the hydrologic system, vegetation, biodiversity, and ecosystem services. Snow cover classifications based on MCSCD and RAL will be beneficial to assess the dynamic snow cover changes in different regions. Therefore, we will consider expanding the scope of snow cover classifications in the future and provide a reference for targeted monitoring and snow cover disaster prevention in different snow cover areas.

## 4. Conclusions

Using the NIEER AVHRR product during 1981–2020, we proposed a persistence-based classification of snow cover across China with MCSCD and RAL as criteria. Snow cover classification across China was divided into persistent snow cover, periodic variable snow cover, and non-periodic variable snow cover. Non-periodic variable snow cover can be divided into two types: fixed-type and ephemeral-type. Persistent snow cover areas were mainly located in northern Xinjiang, northeast China, central, and eastern Inner Mongolia, and the high-mountain Tibetan Plateau. Most of the Tibetan Plateau, the southern Tianshan Mountains, and parts of northeast China belonged to periodic variable snow cover. The eastern and southern hinterland of the Tibetan Plateau was classified as the ephemeral-type snow cover area. The rest of China was the fixed-type snow cover area.

The extents of persistent and periodic variable snow cover reached their maximum in the 2000s. Since then, the extent of non-periodic variable snow cover had increased significantly during the 2010s due to a reduction in extents in persistent and periodic variable snow cover areas. It is worth noting that ephemeral-type snow cover in the eastern Tibetan Plateau and the southern margin of snow cover areas across China have increased strongly in the past 20 years.

There were great differences in snow properties of snow cover classification areas. In general, snow depth was deeper and SCD was longer in persistent snow cover areas than in other snow cover areas. In contrast, snow depth was shallow and SCD was shorter in non-periodic variable snow cover areas. There are snow disaster risks in all snow cover classification areas. Due to the snow accumulation timing being fixed, snow disasters in persistent and periodic variable snow cover areas are easy to prevent. However, we should pay attention to the disasters in ephemeral-type snow cover areas because of the unpredictable time of snow cover occurrence.

Although a new method is used to analyze the distribution and variations of snow cover classification across China, there are still methodological gaps and uncertainties in this study that need to be further improved in the future: (1) firn on glaciers is not identified from retrieving, thus glaciers are included in the recent snow cover classification, not just seasonal snow. (2) The RAL value is not constant, which depends on the study area and the time series. Therefore, there is uncertainty in the determination of non-periodic variable snow cover types. Because of these two issues, the snow cover classification method based on stability and persistence still needs to be improved. In addition, we will take into account the disaster and benefit factors of snow cover to divide snow cover function areas.

**Author Contributions:** H.L. wrote the original manuscript; X.Z. inspired the main idea, edited the manuscript, and gave useful suggestions; L.Z. and X.H. contributed to processing and analyzing data; J.W. contributed to the investigation and project administration; J.Z. gave many constructive suggestions. All authors have read and agreed to the published version of the manuscript.

**Funding:** This work was funded by the Science & Technology Basic Resources Investigation Program of China (2017FY100503), the Open Foundation from National Cryosphere Desert Data Center (E01Z790205), and the National Natural Science Foundation of China (41861049).

**Institutional Review Board Statement:** Not applicable.

**Informed Consent Statement:** Not applicable.

**Data Availability Statement:** The NIEER AVHRR SCE Product is available at http://www.ncdc.ac.cn/portal/metadata/392be836-6b3b-43d8-a586-7a5474a743bd (accessed on 7 February 2022). The data that support the findings of this study are available on request from the corresponding author. The data are not publicly available due to restrictions on distributing the data.

**Acknowledgments:** We would like to thank two anonymous reviewers and the academic editor, Juraj Parajka, for their very insightful and constructive comments and suggestions, which helped to improve the manuscript substantially.

**Conflicts of Interest:** The authors declare no conflict of interest.

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
