# Peer review of "Classification of Snow Cover Persistence across China"

_water, doi:10.3390/w14060933_

Round 1

Reviewer 1 Report

The work titled, "Snow Cover Classification and Its Variations across China" is a good piece of research as well as a good concept indeed to assess the climate variability in China.  The use of snow classification for monitoring and managing landslides is indeed a welcome approach. 

However, there are some methodological issues that the authors need to clarify and include in the manuscript. 

Ln 83-86: How accurate is snow cover extracted from Landsat-5 TM? and which threshold value has been used. The authors need to clarify it in this section alone and add uncertainty estimates as well.

Moreover, NDSI threshold values depend on topography, vegetation, etc. have the author considered this aspect while evaluating the NDSI. The uncertainty analysis would be helpful for the readers.

Ln 107-109:

The authors need to Justify the criteria of classification in more detail (Why 30 days, or 10 days?. The basis of the classification has to be described in detail, as this is the basis for the whole study.

Ln 175-176:

Authors are suggested to support their findings with the graphical representation of decreasing snowfall, increasing temperature, and other weather parameters. Authors already have this data with them, just to be presented. This can be then supported by the references cited.

The authors are advised to add and discuss recent works from other parts of the world particularly from the Himalayas and suggest how this work can be useful there also.

A separate section regarding the research gaps and uncertainties needs to be added in the manuscript to provide a more balanced approach to concluding scientific findings. Research is never Yes or No!

Reviewer 2 Report

The manuscript is devoted to studying the snow season length, the thickness of the snow cover, and the classification of the territory of China according to various parameters of the snow cover. The results are interesting and may be important in studying the water availability for agriculture and other anthropogenic activities, especially in areas with a lack of precipitation during the warm period and with respect to climate change. However, several issues need to be corrected before publication.

The title or keywords should reflect the use of remote sensing data.

The introduction lacks an analysis of approaches used outside of China to classify snow cover. For example, what methods, indicators, and snow parameters are commonly used in other countries, and what are the differences in the approaches of the current study with others? This is useful in substantiating the choice of methods and indices used in the manuscript to reclassify snow data in China.

Line 40: What does "SWE" mean?

Section 2: In what software package are the maps made?

Figure 1: Please add for the red and yellow colors in the legend that this is a non-periodic unstable snow cover.

Lines 119-126: It is helpful to depict some of the names of the geographical objects in Fig. 1.

Line 142: "South of 25°N was mainly a snow-free area." Please provide more details - according to Figure 1, snow cover is presented south of 25°N (i.e., west of 120°E mostly no snow cover about south of 22°N and east of 120°E mostly no snow cover about south of 25°N). There is no discussion of these results.

Lines 179-181: Please also indicate the change in the area of each snow cover class over time. It is imperative to present data on the change in each class of snow cover area in the form of a table (area of each class in the studied decades, change in area in km2 and %).

Figure 3: It needs to additionally show maps of the increase and decrease in the areas of the identified snow cover classes (each decade compared to the previous decade). Otherwise, according to the maps presented in the manuscript, it is challenging to assess the change in the area of each class visually.
Section 3.2: There is no discussion of the results obtained. The probable reasons for the change in the area of each class of snow cover have not been considered, and no evidence has been given of the influence of various factors on the change in the area of each class. The analysis of changes in the precipitation and air temperature in the winter period in the studied years was not carried out.

A grammar check is required (articles are omitted or extra articles are given, etc.).

The manuscript requires major revisions.

Round 2

Reviewer 1 Report

Thanks for incorporating the suggestions.

Author Response

Many thanks for your insightful comments and suggestions on our manuscript.

Reviewer 2 Report

The authors fully responded to my comments and made the required corrections to the text. Therefore, the manuscript can be accepted for publication in its current form.

Author Response

(The authors gave the same response as above.)
